# VideoPrompter: An Ensemble of Foundational Models for Zero-Shot Video Understanding

## Abstract

Vision-language models (VLMs) classify the query video by calculating a similarity score between the visual features and text-based class label representations. Recently, large language models (LLMs) have been used to enrich the text-based class labels by enhancing the *descriptiveness* of the class names. However, these improvements are restricted to the text-based classifier only, and the query visual features are not considered. In this paper, we propose a framework which combines pre-trained discriminative VLMs with pre-trained generative video-to-text and text-to-text models. We introduce two key modifications to the standard zero-shot setting. First, we propose language-guided visual feature enhancement and employ a video-to-text model to convert the query video to its descriptive form. The resulting descriptions contain vital visual cues of the query video, such as what objects are present and their spatio-temporal interactions. These descriptive cues provide additional semantic knowledge to VLMs to enhance their zero-shot performance. Second, we propose class-specific prompts to LLMs to generate more meaningful descriptions to enrich class label representations. Specifically, we introduce prompt techniques to create a Tree Hierarchy of Categories for class names, offering a higher-level action context for additional visual cues, We demonstrate the effectiveness of our approach in video understanding across three different zero-shot settings: 1) video action recognition, 2) video-to-text and text-to-video retrieval, and 3) time-sensitive video tasks. Consistent improvements across multiple benchmarks and with various VLMs demonstrate the effectiveness of our proposed framework. Our code will be made publicly available.

## 1 Introduction

Open-vocabulary models (Roth et al., 2023; Radford et al., 2021; Jia et al., 2021; Yuan et al., 2021) have demonstrated impressive performance in downstream tasks. These models undergo contrastive training on large amounts of image-text pairs, aligning their embeddings in a shared space. However, extending these models to video tasks poses significant challenges mainly for two reasons: (1) due to large computational expenses, and (2) the requirement of gathering large-scale video-text pairs. Therefore, it is critical to effectively leverage pre-trained image-language models for video tasks without affecting their zero-shot capabilities.

To extend pre-trained image-language models to videos, there are two dominant approaches. The first approach takes inspiration from the prompt learning methods (Zhou et al., 2022c;b; Jia et al., 2022) and introduces learnable prompts or adapters to text, vision, or both sides (Yang et al., 2023; Wasim et al., 2023). In contrast, the second approach fine-tunes the whole pre-trained image-language model for video tasks (Rasheed et al., 2023; Luo et al., 2022; Wang et al., 2021; Ni et al., 2022). The aforementioned approaches have several drawbacks, e.g., the introduction of additional learnable parameters that add to overall model complexity or require extensive fine-tuning to adapt the model for video tasks. Further, these methods require access to the true distribution of the target task, which can be prohibitive in test-time adaptation and data-scarce environments.

Recently, a new line of work has emerged (Menon & Vondrick, 2022; Pratt et al., 2022; Novack et al., 2023; Roth et al., 2023) that incorporates large language models (LLMs), such as GPT-3 to provide additional semantic context to existing vision-language models (VLMs) and requires no further training. These methods query LLMs to replace class names with enriched language descriptors in

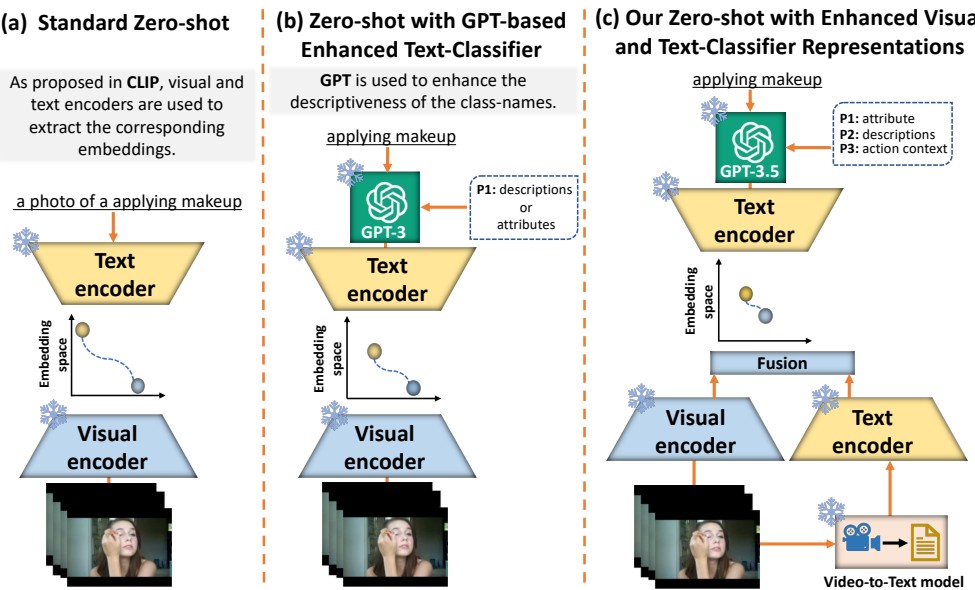

Figure 1: **(a)** The standard pre-training for zero-shot classification (e.g., CLIP (Radford et al., 2021)). **(b)** Existing variants for enhancing zero-shot classification (Pratt et al., 2022; Menon & Vondrick, 2022) using GPT descriptions and attributes that improve text-based classifier features. **(c)** Our proposed framework to enhance both classifier and visual representations. It employs a video-to-text model to generate description of the query video, and these descriptive cues are combined with the visual information. A text-to-text generative model (GPT-3.5) is prompted for class attributes, descriptions, and action context to enhance the class diversity for the text-based classifier.

order to increase class descriptiveness. However, these studies have primarily focused on modifying the text-based classifier only, we propose a twofold improvement approach, simultaneously refining class representations and enriching visual features. Moreover, despite their promising results in image classification, *the applicability of these method in the context of video understanding remains an open question, and our work aims to address this gap*.

We present a framework called VideoPrompter that aims to enhance the test-time zero-shot performance of the existing VLMs for video understanding and introduce two modifications to the standard zero-shot framework. *First,* we query a video-to-text generative model to convert the input video to language representation, as this generated representation contains vital visual cues (*such as what objects are present and how they interact spatially and temporally*), which in turn helps the VLMs to better understand the given video. For instance, for a video shown in Figure 1 (c), the video-to-text model can provide detailed textual description e.g., "*in this video, a woman is seen applying mascara using a makeup brush*". As humans, we can effortlessly leverage such descriptive information and form a visual image of the video's content in our minds. *Second,* to enrich the class representations of the text classifier, we query LLM with class-specific prompts and generate two types of

To summarize, we make the following contributions:

1. We introduce a framework that is an ensemble of video-to-text and text-to-text generative models to increase the zero-shot performance of existing VLMs for video understanding.

2. We introduce class-specific prompts to enrich the classifier representations and also propose a novel way to generate high-level action contexts from the dataset.

3. Our framework offers a plug-and-play module adaptable to various existing VLMs such as CLIP (Radford et al., 2021), ViFi-CLIP (Rasheed et al., 2023), Action-CLIP (Wang et al., 2021), and AIM (Yang et al., 2023).

4. We demonstrate results on three different video settings namely: action recognition, video-to-text, and text-to-video retrieval, and time-aware video tasks, and show results and ablation on 7 different datasets.

## 2 VIDEOPROMPTER

### 2.1 OVERVIEW

Let $x$ denote the query video and $C$ denote the target categories. Let $f_V$ and $f_T$ respectively be the visual and text encoders of a VLM such as CLIP. The zero-shot video classification can be defined as nearest neighbor retrieval as follows:

$$\tilde{c} = \arg\max_{c \in C} \; \cos\left(f_V(x), f_T(p(c))\right), \tag{1}$$

with prompt $p(c) = $ A photo of a {c}, *cos* represents cosine similarity between visual and textual features.

The overall framework of VideoPrompter is presented in Figure 1 (c) . *The input query video $x$ is passed through a video-to-text conversational model, referred to here as $F_\Theta$ to generate corresponding video textual description.* This video textual description is then processed by the VLM text encoder, $f_T$, and fused together with the video features to get the enriched visual features $\tilde{f}_V$. On the classifier-side, an LLM model, denoted by $F_\phi$, converts the target categories $C$ to corresponding language attributes and descriptions. These class label descriptions are then processed by the VLM text encoder, $f_T$, to enhance class label features, $\tilde{f}_T$. Our proposed zero-shot classification can then be defined as follows:

$$\tilde{c} = \arg\max_{c \in C} \; \cos(\tilde{f}_V, \tilde{f}_T|_c). \tag{2}$$

In the following sections, we examine how the visual features $\tilde{f}_V$ and the enriched class representations $\tilde{f}_T$ are derived. For clarity, the descriptions generated by the video-to-text model (Maaz et al., 2023) are referred to as video textual descriptions. On the other hand, attributes and descriptions generated by LLM (Brown et al., 2020) are called language attributes and language descriptions.

### 2.2 VIDEO-TO-TEXT GUIDED VISUAL FEATURE ENHANCEMENT

Given a video-language model, we aim to enhance its zero-shot performance by employing a video-to-text conversational model, i.e., Video-ChatGPT (VGPT) (Maaz et al., 2023), to analyze the video content. We prompt VGPT as "*describe the activity in the video*". VGPT leverages its spatiotemporal alignment between BLIP (Li et al., 2022) and an LLM (Chiang et al., 2023) to capture temporal dynamics and frame-to-frame consistency relationships, allowing it to generate coherent video textual descriptions of the events unfolding in the video. These video textual descriptions embed vital spatial and temporal information about the video events. The generated video textual descriptions are passed through the VLM text encoder $f_T$ to generate a video description-level embedding.

Since the VLMs share a common image-text embedding space due to their contrastive learning objective (Radford et al., 2021), we found a simple weighted average of the video embedding and video textual description embedding to be efficient, as shown by fusion in Figure 1 (c). The enhanced visual embedding $\tilde{f}_V$ is given as:

$$\tilde{f}_V = \beta_1 \cdot f_V(x) + \beta_2 \cdot f_T\left(F_\Theta(x)\right), \tag{3}$$

where $\beta_{1\sim2}$ denotes the weights for the query video and video textual description embeddings, respectively.

### 2.3 TEXT-TO-TEXT GUIDED CLASSIFIER REFINEMENT

In our approach, we leverage an LLM, (Brown et al., 2020) and generate class-specific language descriptors. We found that class-specific descriptors can better adapt to the action-recognition datasets, and unlike (Pratt et al., 2022), only a small number of descriptors are required. For example, our framework only requires three descriptors, unlike 50 for (Pratt et al., 2022) in the case of the UCF-101 dataset (Soomro et al., 2012) 3.1.4. Furthermore, we arrange video action datasets in high-level

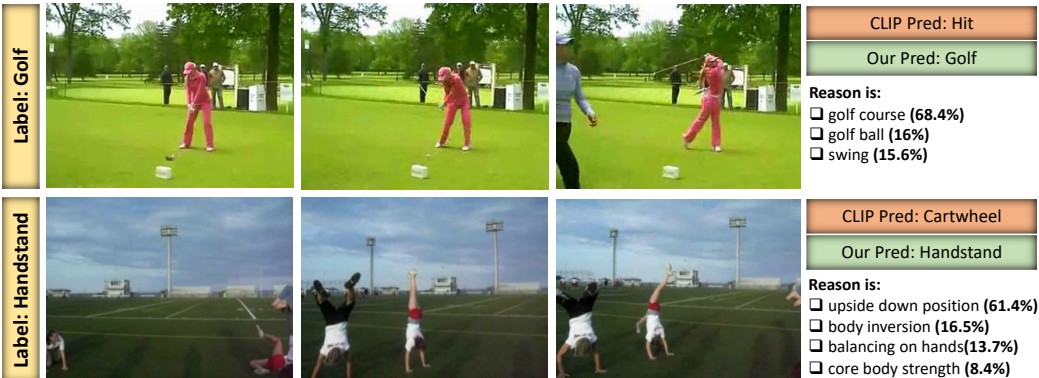

Figure 2: VideoPrompter's Interpretability. We divide our proposed attributes (section 2.3) and get the cosine similarity between the individual attribute and query video to find the contribution of each attribute. For example, the top row shows the prediction of our VideoPrompter is "golf" and the reason the model made this decision is because of *golf-course* in the background, *golf-ball* in the scene, and *swing* of the golf-stick. Similarly, the bottom row shows the *upside down position* played a vital role in the model to make the decision that it's a "handstand" action.

action contexts. For instance, all sports-related actions (basketball, cricket, baseball) can be added under one high-level action context, i.e., "playing sports". We propose to employ LLM to exploit this property of action-recognition datasets and generate high-level action context to provide additional cues to the classifier. In summary, we extract class-specific language descriptions (Sec. 2.3) from an LLM with high-level action context (Sec. 2.3.2).

### 2.3.1 CLASS-SPECIFIC LANGUAGE DESCRIPTORS

For action recognition, we propose to leverage an LLM, which is GPT-3.5 in our case (Brown et al., 2020) to take a class name as input and generate two types of class-specific language descriptors:

1. **Language Attributes** offer object-level visual cues to encourage the classifier to employ these features instead of just the class names. For instance, for the video action of "baby-crawling" the generated language attributes are: *baby, crawling, hand, knees*. We prompt GPT-3.5 as:

   ```
   Q: What are the distinct visual characteristics to identify a
   {class-name} video action?
   ```

2. **Language Descriptions** describe how a specific action is performed. It includes step-by-step directions for that action to facilitate the model's comprehension of the temporal context. For instance, for the video action of "baby-crawling" the generated language description is: *a baby is seen on all fours, using their arms and knees to move across the floor. They alternate their arms and legs in a crawling motion, exploring their surroundings with curiosity*. We prompt GPT-3.5 as:

   ```
   Q: How {class-name} action is performed visually?
   ```

The text encoder $f_T$ extracts the embeddings of such generated language attributes and language descriptions. The modified class label representation $\tilde{f}_T$ is the average of the embeddings of language attributes and language descriptions.

In problem setting of video-to-text or text-to-video retrieval, each video is paired with a corresponding caption. We employ GPT-3.5 to take the input caption and generate informative and semantically similar captions. For instance, for an input caption, "man is giving a review on a vehicle", the generated caption is "*a person provides feedback on a car*". Similarly, for an input caption "baseball player hits ball", the generated caption is "*the ball is hit by a player in a baseball game*". We prompt GPT-3.5 as:

Table 1: A short summary of high-level action context for HMDB-51, UCF-101, and K400 datasets.

| Dataset | High-Level Action-Context |
|---------|---------------------------|
| HMDB-51 | self-grooming, physical activities, sports, eating, social interactions, artistic activities, other |
| UCF-101 | self-grooming, playing music, playing sports, exercise and fitness, water activities, household chores, creative activities, other |
| K400 | self-grooming, playing music, playing sports, exercise and fitness, household chores, social interactions, creative activities, transportation activities, water activities, other |

```
Q: Given a caption: {input caption}, generate a visually similar
captions.
```

The text encoder $f_T$ extracts the embeddings of the generated captions. The modified class label representation $\tilde{f}_T$ is the average of the embeddings of the original and generated captions.

### 2.3.2 INTEGRATING HIGH-LEVEL VIDEO ACTION CONTEXT

We proposed a novel way of querying LLMs and divide the dataset into various *high-level action contexts* such that all video-action classes semantically close to each other are grouped under one high-level action context. We prompt - GPT-3.5 as:

```
Q: Divide the list of {class-names} into parent and child classes.
Such that actions that are visually similar to each other are in
the same group. If the action is not similar to any other action
in the list, assign it to others.
```

As a result, this converts the standard input prompt (*a photo of a {class-name}*) to high-level action context prompt: *a photo of a {high-level context} i.e. {class-name}*. For instance, for the video action of "*drumming*", the prompt becomes "*a photo of a playing music i.e., drumming*". The text encoder $f_T$ extracts the embeddings of the generated high-level action context prompt, which can be averaged with the embeddings of the language attributes and descriptions to create a context-based classifier. A summary of high-level action context for the different datasets is provided in Table 1, while the comprehensive list of Tree Hierarchy of Categories is provided in the Appendix B C, D.

## 3 EXPERIMENTAL PROTOCOLS

We evaluate the effectiveness of VideoPrompter under three different video zero-shot settings: a) action recognition, b) text-to-video and video-to-text retrieval, and c) time-sensitive video tasks (Bagad et al., 2023). Additionally, we demonstrate that our VideoPrompter provides interpretability of the model decisions (Figure. 2). We use the ViT-B/16 backbone and sparsely sample 32 frames as consecutive frames are highly redundant with single-view evaluation (Rasheed et al., 2023). In the case of CLIP, the video embedding is obtained by averaging the frame-level embeddings, (Portillo-Quintero et al., 2021; Rasheed et al., 2023), while for other methods we follow their default settings. The video-adapted models (Rasheed et al., 2023; Wang et al., 2021; Yang et al., 2023) are pre-trained on the K400 dataset (Kay et al., 2017).

We use three video textual descriptions from VGPT and only two language descriptors: language attributes and descriptions. To make sure that all three video textual descriptions generated by VGPT are diverse, we set its temperature (likelihood of selecting lower probability tokens) to 0.5., while for GPT-3.5, as we only prompt the model once for each descriptor, we set its temperature to 0.2 to generate more focused and deterministic descriptors. The video textual descriptions generated by VGPT are trimmed to be consistent within the context length of the text encoder (Radford et al., 2021), and we also apply CLIP-based filtering as a pre-processing step, discussed in 3.1.2 to remove erroneous video textual descriptions. We set $\beta_1$ equal to 1.0, and $\beta_2$ is calculated as cosine-similarity between the embeddings of query video and its video textual description. This ensures that a video textual description that is consistent with the query video is given higher weight. As AIM (Yang

Table 2: Zero-shot action recognition (top-1 %) using our VideoPrompter provides consistent improvements across different VLMs and video datasets.

| Method | VideoPrompter | HMDB | UCF | SSv2 | K400 |
|---|---|---|---|---|---|
| *Uni-modal zero-shot action recognition models* | | | | | |
| ASR (Wang & Chen, 2017) | – | 21.8 | 54.4 | – | – |
| ZSECOC (Qin et al., 2017) | – | 22.6 | 15.1 | – | – |
| UR (Zhu et al., 2018) | – | 24.4 | 17.5 | – | - |
| E2E (Brattoli et al. (2020)) | – | 32.7 | 48 | – | – |
| ER-ZSAR Chen & Huang (2021) | – | 35.3 | 51.8 | – | – |
| *Adapting pre-trained image VL models* | | | | | |
| XCLIP (Ni et al. (2022)) | – | 44.6 | 72.0 | – | – |
| A5 (Ju et al. (2022)) | – | 44.3 | 69.3 | – | – |
| CLIP (Radford et al., 2021) | ✗ | 37.5 | 61.72 | 2.72 | 44.53 |
| | ✓ | $50.79_{(+13.29)}$ | $72.77_{(+11.05)}$ | $4.87_{(+2.15)}$ | $49.17_{(+4.64)}$ |
| VIFI-CLIP (Rasheed et al., 2023) | ✗ | 51.82 | 77.5 | 4.5 | – |
| | ✓ | $57.12_{(+5.30)}$ | $79.56_{(+2.06)}$ | $5.40_{(+0.87)}$ | – |
| AIM (Yang et al., 2023) | ✗ | 51.27 | 72.19 | 4.01 | – |
| | ✓ | $54.37_{(+3.10)}$ | $78.50_{(+6.31)}$ | $5.84_{(+1.83)}$ | – |
| ActionCLIP (Wang et al., 2021) | ✗ | 49.20 | 69.52 | 4.42 | – |
| | ✓ | $51.65_{(+2.45)}$ | $77.07_{(+7.55)}$ | $5.27_{(+0.85)}$ | – |

Table 3: Our VideoPrompter boosts zero-shot Text-to-Video and Video-to-Text Retrieval performance.

| Method | VGPT | GPT3.5 | Video-to-Text | | Text-to-Video | |
|---|---|---|---|---|---|---|
| | | | R@1 | R@5 | R@1 | R@5 |
| CLIP (Radford et al., 2021) | – | – | 28.19 | 52.90 | 31.7 | 54.0 |
| Video-CLIP (Xu et al., 2021) | – | – | 30.6 | – | 10.4 | 22.2 |
| FrozenInTime (Bain et al., 2021) | – | – | – | – | 24.7 | 46.9 |
| CLIP4CLIP (Luo et al., 2022) | – | – | – | – | 32.0 | **57.0** |
| CLIP (Radford et al., 2021) | ✓ | ✗ | 30.59 | 53.90 | 32.8 | 54.5 |
| | ✓ | ✓ | $31.30_{(+3.11)}$ | $55.10_{(+2.2)}$ | $33.50_{(+1.8)}$ | $56.49_{(+2.49)}$ |

et al., 2023) only comprises a visual encoder, we remove its clasification layers and employ vanilla-CLIP text encoder for zero-shot analysis.

**Video Action Recognition.** Our VideoPrompter achieves consistent improvements across various models and benchmarks, as shown in Table 2. We observe that when CLIP is incorporated within our framework, it performs on par with the fully-finetuned methods like ViFi-CLIP and ActionCLIP and outperforms adapter-based method AIM. *Effectiveness of High-Level Action Context in Video Action Recognition:* We observe that for datasets like UCF-101 (Soomro et al., 2012), HMDB-51 (Kuehne et al., 2011) and K400 (Kay et al., 2017), having diverse contexts, our proposed method finds effective and natural high-level action contexts and boost the performance across all methods, shown in Table 6. However, for datasets like SSv2 (Goyal et al., 2017), where class names highly correlate to each other such as *letting something roll along a flat surface* and *letting something roll down a slanted surface*, we found GPT-3.5 divides all of the actions in one class "manipulating objects" and we observed no improvement with high-level action context.

**Text-to-Video and Video-to-Text Retrieval.** We present recall at a rank (R@K, where k = 1,5) on 1k-A split-set of the MSR-VTT (Xu et al., 2016) dataset with the CLIP model. In the retrieval setting, for each video, we obtain 10 video textual descriptions using VGPT, and, for every caption we generate two more semantically similar but informative captions using GPT-3.5. As shown in Table 3 , our method consistently increases the performance.

**Time-Sensitive Video Tasks.** Bagad et al. (2023) discussed time awareness in video models and showed that the recent VLMs struggle to understand simple temporal relations such as *before and after*. To show that our VideoPrompter can increase the time-awareness of the existing VLMs, we report the time-consistency score on the before/after synthetic dataset (details of the dataset provided in Appendix A) (Bagad et al., 2023) and time-aware setting of Charades dataset (Sigurdsson et al., 2016). For each query video, we obtain 10 video textual descriptions using VGPT. In a time-aware setting, considering the nature of the problem, language attributes and descriptions are not used. Our framework enhances the performance on both benchmarks (Table 4). We obtain a substantial gain of 10%, even without using the language attributes and descriptions, which shows that our framework can provide additional cues to the VLMs to understand temporal relations.

Table 4: Our VideoPrompter increases the time awareness in VLMs. SD shows synthetic dataset.

| Method | Time-consistency score | |
| --- | --- | --- |
| | SD | Charades |
| CLIP (Radford et al., 2021) | 50.0 | 56.0 |
| Video-CLIP (Xu et al., 2021) | 51.1 | 47.1 |
| CLIP4Clip (Luo et al., 2022) | 51.1 | - |
| CLIP2Video (Fang et al., 2021) | 47.8 | - |
| CenterCLIP (Zhao et al., 2022) | 46.1 | - |
| VindLU (Cheng et al., 2023) | 52.0 | - |
| Frozen in Time (Bain et al., 2021) | 50.0 | - |
| VideoPrompter (CLIP) | 60.0 (+10) | 57.4 (+1.4) |

Table 5: CUPL generates 50 descriptions for each class. VideoPrompter outperforms CUPL with only 3 language descriptors and a video textual description.

| Method | GPT | VGPT | HMDB-51 | UCF101 | SSv2 | Prompts |
| --- | --- | --- | --- | --- | --- | --- |
| CLIP | - | - | 37.5 | 61.72 | 2.72 | - |
| CUPL | ✓ | ✗ | 49.14 | 73.67 | 4.06 | 50 |
| CUPL | ✓ | ✓ | 50.44 | 73.54 | 4.81 | 50 |
| VideoPrompter | ✓ | ✓ | 52.51 (+3.37) | 73.88 (+0.21) | 4.87 (+0.81) | 3 |

Table 6: Action context to further aid action-recognition.

| Method | VideoPrompter | Action-Context | HMDB | UCF | K400 |
| --- | --- | --- | --- | --- | --- |
| CLIP (Radford et al., 2021) | ✗ | ✗ | 37.5 | 61.72 | 44.53 |
| CLIP (Radford et al., 2021) | ✓ | ✗ | 50.79 | 72.77 | 49.17 |
| CLIP (Radford et al., 2021) | ✓ | ✓ | 52.51 (+15.01) | 73.88 (+12.16) | 52.03 (+7.50) |
| ViFi-CLIP (Rasheed et al., 2023) | ✗ | ✗ | 51.82 | 77.5 | - |
| ViFI-CLIP (Rasheed et al., 2023) | ✓ | ✗ | 57.12 | 79.56 | - |
| ViFI-CLIP (Rasheed et al., 2023) | ✓ | ✓ | 57.94 (+6.12) | 80.70 (+3.2) | - |
| AIM (Yang et al., 2023) | ✗ | ✗ | 51.27 | 72.19 | - |
| AIM (Yang et al., 2023) | ✓ | ✗ | 54.37 | 78.50 | - |
| AIM (Yang et al., 2023) | ✓ | ✓ | 55.54 (+4.27) | 77.90 (+5.71) | - |
| Action-CLIP (Wang et al., 2021) | ✗ | ✗ | 49.20 | 69.52 | - |
| Action-CLIP (Wang et al., 2021) | ✓ | ✗ | 51.65 | 77.07 | - |
| Action-CLIP (Wang et al., 2021) | ✓ | ✓ | 54.50 (+5.3) | 77.47 (+7.95) | - |

## 3.1 UNDERSTANDING DIFFERENT COMPONENTS OF VIDEOPROMPTER

In this section, we study different components of our framework. We use the CLIP model with ViT-B/16 visual encoder in all our ablations.

### 3.1.1 DESIGN CHOICES

To study the design effectiveness of our framework, we discuss various other design choices. First, to show that a combination of query video and its video textual description embedding is the optimal choice, we remove the visual encoder and only take the similarity of the video textual description embedding with the class representations, as shown in Figure 3 (left). We observe that both modalities (visual information and corresponding descriptive information) complement each other and removing visual embedding leads to sub-optimal results.

Further, we also study the impact of removing either the video-to-text model (VGPT) or the text-to-text model (GPT-3.5), as shown in Figure 4. While employing these modules individually results in improved performance across all four benchmarks, their combination exhibits a complementary relationship delivering the most optimal performance.

We also examined the possibility of predicting class names directly by providing GPT-3.5 with video textual descriptions and instructing it to select the closest matching class from a predefined list, we found that this approach fell short of producing optimal results.

### 3.1.2 FILTERING OF VIDEO TEXTUAL DESCRIPTIONS

We apply CLIP-based filtering as a pre-processing step to remove erroneous visual textual descriptions. Specifically, we generate 10 visual textual descriptions for each query video and extract corresponding textual embeddings along with the visual embedding of the query video. Cosine similarity between these embeddings is taken to filter the top-3 visual textual descriptions. As shown in Figure 3 (middle), filtering of visual textual descriptions further increases the performance of our framework. We only apply filtering in the action-recognition setting.

### 3.1.3 IMPACT OF VISUAL TEXTUAL DESCRIPTION DIVERSITY

In order to analyze how the diversity of visual textual descriptions impacts our framework, we experimented with two varying temperature settings (0.2 and 0.5). These temperature settings directly influence the probability of selecting less common tokens, thereby increasing the diversity of the

generated descriptions. As shown in Figure 3 (right), the higher temperature setting leads to better results (as it generates more diverse descriptions). Here, *VGPT only* indicate that only video textual descriptions are used, while language attributes and descriptions are not employed.

### 3.1.4 COMPARISON WITH CUPL

We compare our work with one of the recent works CUPL (Pratt et al., 2022) and show that only a handful of carefully designed video prompts combined with the video-to-text guided visual feature enhancement module leads to superior performance. CUPL employs GPT-3 and designs multiple dataset-specific prompts to generate the language descriptions. For instance, for UCF-101, (Pratt et al., 2022) design 5 prompts and generate 10 responses for each prompt leading to 50 descriptions in total. As shown in Table 5, our framework obtains superior performance with only 3 language descriptors i.e. language attributes, language descriptions, and high-level action context. Further, recent work (Roth et al., 2023) found descriptor ensemble as the main driver for performance in the case of a large number of prompts and showed comparable performance with randomized descriptors. This further validates our work that only a few carefully designed video prompts are enough to enrich the class representations.

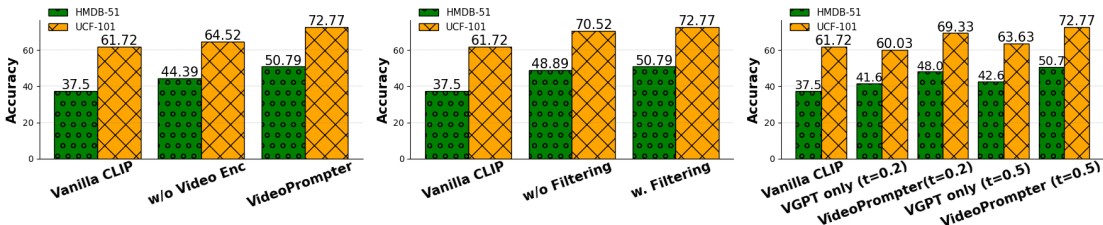

Figure 3: **(left)** Combination of embeddings of query video and its video textual description leads to optimal choice. **(middle)** CLIP-based filtering is applied as a pre-processing step, it further boosts the performance by removing the erroneous video textual descriptions. **(right)** A higher temperature setting leads to better results, as it generates more diverse video textual descriptions.

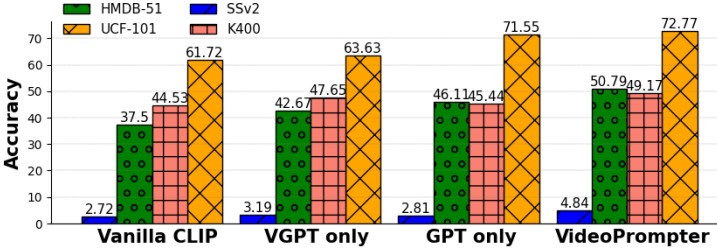

Figure 4: Video-textual description and language descriptors complement each other and their combination (VideoPrompter) leads to optimal results. We found this behavior consistent across all benchmarks and models.

## 4 LITERATURE REVIEW

**Vision-Language Models.** VLMs (Radford et al., 2021; Jia et al., 2021) have shown impressive generalization capabilities on various downstream tasks including open-vocabulary image recognition (Zhang et al., 2021; Zhou et al., 2022c;b), object detection (Gu et al., 2021; Rao et al., 2022; Zhou et al., 2022d) and image segmentation (Ding et al., 2022; Zhou et al., 2022a). However, the extension of these models to video-related tasks poses significant challenges, primarily due to the substantial computational costs and the requirement to collect large-scale video-text pairs. Therefore, various methods have been proposed to effectively leverage pre-trained image-language models for video tasks (Rasheed et al., 2023; Wang et al., 2021; Yang et al., 2023; Wasim et al., 2023). Nevertheless, these approaches either introduce additional learnable parameters that add to overall model complexity or require extensive fine-tuning to adapt the model for video tasks. Further, these methods require access to the true distribution of the target task, which can be prohibitive in test-time adaptation and data-scarce environments.

**Improving Text Classifier Representations Using LLMs.** As open-vocabulary models classify the input by calculating a similarity score between the image/video and textual prompt (a photo of a {class-name}.) for each class, this makes the models' performance directly dependent on the *descriptiveness* of the class names. Recently, a new line of work has emerged (Menon & Vondrick, 2022; Pratt et al., 2022; Roth et al., 2023; Novack et al., 2023) that incorporates LLMs to enrich these class names and requires no further training or access to the true distribution of the target task. (Menon & Vondrick, 2022) highlights that language can provide additional context to the classifier, and LLM is employed to describe visual features that distinguish that object in an image. (Pratt et al., 2022) further explores this idea and uses LLM to generate multiple descriptors for each class name in the dataset. For instance, for UCF-101, (Pratt et al., 2022) designed 5 prompts and generated 10 responses for each prompt leading to 50 descriptions in total. Despite the promising results in image classification, *the applicability of these methods (Menon & Vondrick, 2022; Pratt et al., 2022; Roth et al., 2023; Novack et al., 2023) in the context of video understanding remains an open question, and our work aims to address this gap*. Moreover, as these methods make use of LLMs to increase the descriptiveness of the class names, the visual side of the VLMs remains unaltered. Our framework - VideoPrompter - simultaneously refines class representations and enriches visual features, utilizing text-to-text (Brown et al., 2020) and video-to-text models (Maaz et al., 2023) respectively. We do so in a two-step approach: first, we query a video-to-text generative model to convert the input video to language representation (description) and fuse it with the query video embedding to enhance the overall visual representation. Second, we use class-specific prompts to query LLM and generate language descriptors to enrich the class representations.

**Context-based Classifier Enhancement.** (Roth et al., 2023; Novack et al., 2023) showed that furnishing contextual information to the VLM can significantly aid the model in directing its attention to relevant features and help resolve class ambiguities. (Roth et al., 2023) employed LLM to find the higher-level commonalities in the dataset and GPT-3 is used to extract common context from the datasets. For instance, in CUB200-2011, a bird-related dataset,(Roth et al., 2023) generates the context "bird." Likewise, in the eurosat dataset of satellite images, (Roth et al., 2023) outputs the context "land use". However, video-action recognition datasets (Kay et al., 2017; Kuehne et al., 2011; Soomro et al., 2012) generally comprise actions with diverse contexts. For example, the UCF-101 dataset can be subdivided into the following high-level contexts: *self-grooming, playing music, playing sports, exercise and fitness, water activities, household chores, and other activities*, therefore, (Roth et al., 2023) cannot be directly applied to such diverse datasets. Recently, (Novack et al., 2023) proposed a sub-division of the classes to one level lower, i.e., fine-grained classes. However, their method is only applicable to datasets (unlike action recognition) where the classes are not fine-grained. *We proposed an alternative way of querying LLMs and divide the dataset into various high-level action contexts such that all video-action classes semantically close to each other are grouped under one high-level action context*. For instance, all sports-related actions (basketball, cricket, baseball) can be added under one high-level action context, i.e., "playing sports". This high-level action context provides additional cues to further enrich the class representations.

## 5 CONCLUSION

In this work, we introduced a framework - VideoPrompter - to boost the zero-shot performance of existing VLMs for video understanding. We present a systematic way to prompt pre-trained generative video-to-text and text-to-text models to provide additional semantic context to enrich visual and class representations simultaneously. We demonstrate that, without further training, VideoPrompter performs on par with the various existing fully fine-tuned methods and outperforms adapter-based methods. We also discuss various design choices and demonstrate that both video-to-text and text-to-text models complement each other and result in optimal performance. We also introduce a Tree Hierarchy of Categories for class names, offering a higher-level action context for additional visual cues. VideoPrompter achieved consistent improvement across three different zero-shot settings: video action recognition, video-to-text, text-to-video retrieval, and time-sensitive video tasks across multiple benchmarks and models.

## REPRODUCIBILITY

We used GPT-3.5, CLIP, and VideoChatGPT in our work. All of these models/weights/APIs are publicly available. Additionally, the datasets utilized are also publicly available. While details to reproduce our work are provided in Section 2, by following the provided instructions, our experiments can be replicated. We will also release the code upon publication.

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

## A    TIME-AWARE SYNTHETIC DATASET

The dataset (Bagad et al., 2023) contains 180 video-text pairs with different shapes and colors and two types of temporal relationships: *before and after*. Each video has a corresponding correct caption referred to as an *attractor* and a negative caption with flipped events referred to as *distractor*. In the ideal scenario, the VLM should be able to associate query video with the attractor.

## B    HIGH-LEVEL ACTION CONTEXT FOR HMDB-51

**Self-grooming:** brush hair.
**Physical activities:** cartwheel, climb, climb stairs, dive, flicflac, hand stand, jump, pullup, pushup, ride bike, ride horse, run, situp, somersault, stand, swing baseball, talk, turn, walk.
**Sports:** catch, dribble, golf, hit, kickball, kick, shootball.
**Eating:** chew, drink, eat.
**Social interactions:** clap, hug, kiss, shake hands, smile, wave.
**Artistic activities:** draw sword, sit, smoke, fencing, laugh, pour, pick, shoot gun, shoot bow, sword exercise, throw, sword.
**Others:** fall floor, push, punch.

## C    HIGH-LEVEL ACTION CONTEXT FOR UCF101

**Self-grooming:** apply eye makeup, apply lipstick, blow dry hair, brushing teeth, haircut, head massage, shaving beard.
**Playing music:** drumming, playing cello, playing daf, playing dhol, playing flute, playing guitar, playing piano, playing sitar, playing tabla, playing violin.
**Playing sports:** archery, balance beam, band marching, baseball pitch, basketball, basketball dunk, bench press, biking, billiards, bowling, boxing punching bag, boxing speed bag, cricket bowling, cricket shot, field hockey penalty, frisbee catch, gold swing, hammer throw, horse race.
**Exercise and fitness:** body weight squats, handstand pushups, pull ups, lunges, handstand walking, high jump, push ups, wall pushups.
**Water activities:** breast stroke, cliff diving, diving, kayaking, rafting.
**Household chores:** cutting in kitchen, mixing, mopping floor.
**Creative activities:** knitting, typing, writing on board, yoyo.
**Other:** baby crawling, blowing candles, hula hoop, nunchucks, parallel bars, pizza tossing, rope climbing, salsa spin, swing, tai chi, walking with dog, clean and jerk, fencing, front crawl, floor gymnastics, hammering, juggling balls, jump rope, jumping jack, pommel horse, punch, sky diving.

## D    HIGH-LEVEL ACTION CONTEXT FOR K400

**Self-grooming:** applying cream, brushing hair, brushing teeth, cutting nails, dying hair, fixing hair, filling eyebrows, getting a haircut, getting a tattoo, grooming dog, grooming horse, massaging back, massaging feet, massaging legs, massaging person's head, shaving head, shaving legs, shining shoes, trimming or shaving beard, waxing back, waxing chest, waxing eyebrows, waxing legs.
**Playing music:** air drumming, beatboxing, playing accordion, playing bagpipes, playing bass guitar, playing cello, playing clarinet, playing controller, playing didgeridoo, playing drums, playing flute, playing guitar, playing harmonica, playing harp,

playing ice hockey, playing keyboard, playing organ, playing piano, playing recorder, playing saxophone, playing trombone, playing trumpet, playing ukulele, playing violin, playing xylophone, strumming guitar, tapping guitar.

**Playing sports:** archery, arm wrestling, bobsledding, bowling, capoeira, cartwheeling, cheerleading, climbing a rope, climbing tree, contact juggling, disc golfing, dodgeball, drop kicking, golf chipping, golf driving, golf putting, high jump, high kick, hitting baseball, hockey stop, hopscotch, hurdling, hurling (sport), ice climbing, javelin throw, kitesurfing, long jump, paragliding, parkour, passing American football (in game), passing American football (not in game), picking fruit, playing basketball, playing cricket, playing kickball, playing squash or racquetball, playing tennis, playing olleyball, pole vault, riding mechanical bull, riding mountain bike, roller skating, shooting basketball, shooting goal (soccer), shot put, skiing (not slalom or crosscountry), skiing crosscountry, skiing slalom, skydiving, slacklining, sled dog racing, snowboarding, snowkiting, snowmobiling, somersaulting, spinning poi, springboard diving, swing dancing, sword fighting,tobogganing, trapezing, triple jump, vault, wrestling.

**Exercise and fitness:** abseiling, bench pressing, blasting sand, blowing leaves, blowing nose, blowing out candles, bouncing on trampoline, braiding hair, bungee jumping, carrying baby, chopping wood, clapping, climbing ladder, crawling baby, crossing river, crying, digging, exercising arm, exercising with an exercise ball, extinguishing fire, feeding birds, feeding fish, feeding goats, front raises, garbage collecting, gargling, hammer throw, holding snake, jogging, jumping into pool, laughing, laying bricks, lunge, mopping floor, moving furniture, mowing lawn, opening present, planting trees, reading book, rock climbing, running on treadmill, scrambling eggs, shaking hands, shaking head, sharpening pencil, shoveling snow, side kick, situp, skipping rope, smoking, smoking hookah, snatch weight lifting, sneezing, sniffing, squat, stomping grapes, stretching arm, stretching leg, surfing crowd, swinging legs, swinging on something, tai chi, taking a shower, tying bow tie, tying knot (not on a tie), tying tie, using remote controller (not gaming), walking the dog, washing feet, washing hair, washing hands, welding, yawning.

**Household chores:** assembling computer, breading or breadcrumbing, building cabinet, building shed, cleaning floor, cleaning gutters, cleaning pool, cleaning shoes, cleaning toilet, cleaning windows, counting money, cutting pineapple, cutting watermelon, doing laundry, doing nails, folding clothes, folding napkins, folding paper, frying vegetables, ironing, making bed, plastering, reading newspaper, ripping paper, sanding floor, setting table, sharpening knives, shredding paper, sweeping floor, trimming trees, unboxing, using computer, washing dishes, watering plants.

**Social interactions:** answering questions, auctioning, celebrating, checking tires, giving or receiving award, news anchoring, presenting weather forecast, testifying, texting, waiting in line.

**Creative activities:** arranging flowers, balloon blowing, bandaging, bartending, bee keeping, blowing glass, bookbinding, carving pumpkin, country line dancing, cracking neck, drawing, making jewelry, playing cards, playing monopoly, recording music, sticking tongue out, weaving basket, wrapping present, writing.

**Transportation activities:** biking through snow, driving car, driving tractor, flying kite, hoverboarding, motorcycling, riding a bike, riding camel, riding elephant, riding mule, riding

or walking with horse, riding scooter, riding unicycle, unloading
truck, using segway.
**Water activities:** canoeing or kayaking, diving cliff, ice fish-
ing, ice skating, jetskiing, parasailing, sailing, scuba diving,
snorkeling, surfing water, swimming breast stroke, swimming but-
terfly stroke, water skiing, water sliding, windsurfing.
**Other:** baby waking up, bending back, bending metal, brush paint-
ing, catching fish, catching or throwing baseball, catching or
throwing frisbee, catching or throwing softball, changing oil,
changing wheel, clay pottery making, clean and jerk, curling
hair, deadlifting, doing aerobics, dribbling basketball, drink-
ing, drinking beer, drinking shots, drumming fingers, dunking
basketball, egg hunting, flipping pancake, grinding meat, gym-
nastics tumbling, hugging, jumpstyle dancing, kicking field goal,
kicking soccer ball, kissing, marching, petting animal (not cat),
petting cat, playing badminton, playing chess, playing cymbals,
playing paintball, playing poker, pull ups, pumping fist, pumping
gas, punching bag, punching person (boxing), push up, pushing car,
pushing cart, pushing wheelchair, shearing sheep, shuffling cards,
sign language interpreting, ski jumping, slapping, spraying, swim-
ming backstroke, tango dancing, tickling, tossing coin, training
dog.

