# OpenReview forum: "VIDEOPROMPTER: AN ENSEMBLE OF FOUNDATIONAL MODELS FOR ZERO-SHOT VIDEO UNDERSTANDING"
_ICLR.cc/2024/Conference — Submitted to ICLR 2024_

### Official Review · Reviewer_FnrC · 2023-10-26

**Soundness:** 3 good
**Presentation:** 3 good
**Contribution:** 2 fair
**Rating:** 5
**Confidence:** 4

**Summary:**

The manuscript proposes an ensemble framework for video understanding based on pre-trained visual-language models, which involves utilizing LLM to enhance the descriptiveness of text labels and leveraging a video-to-text model to enhance video representations. The authors conducted comprehensive experiments on action recognition, video-text retrieval, and time-sensitive video tasks, demonstrating the effectiveness of the approach in zero-shot scenarios.

**Strengths:**

The experiments and ablation studies are conducted comprehensively, validated on different pre-existing architectures, and taken into account various types of video data.

Employing the video-to-text model (not limited to VGPT and caption models) is novel and worthy to explore in the video field. Fusing the text and video representations is depicted to be beneficial in bridging the gap between video and textual labels in the embedding space.

The manuscript provides a detailed explanation and examples of prompting the GPT to refine the simple textual label, which in turn enhances reproducibility.

**Weaknesses:**

In the 'video-to-text guided visual feature enhancement' (section 2.2), the adopted VGPT relies on CLIP-ViT-L and vicuna, where the computational cost of performing multiple inferences (including text embedding and filtering) far exceeds that of the basic video understanding model. This limits the practical value of the proposed approach.

Except for CLIP, an image-language pre-trained model targeted specifically for the image field, the proposed approach shows relatively limited performance gain in other video-based models (ViFi-CLIP, AIM, ActionCLIP), considering the additional computational requirements.

The configurations of adopted pre-trained models (AIM, ActionCLIP, …) remain unclear, which datasets are these models pre-trained on (e.g. K400, K700, …)? For AIM, do the authors directly remove the classification layers?

**Questions:**

Are the high-level action contexts mentioned in the manuscript manually designed?

---

> ### Author Response · Authors · 2023-11-23
> **Response to Reviewer  FnrC**
>
> Thank you for your detailed review, below we address the key concerns.
>
> **Q1: *The computational cost of performing multiple inferences far
> exceeds that of the basic video understanding model.***
>
> Kindly note that the VideoPrompter offers a flexible framework, allowing
> for the substitution of various modules, including the video-2-text
> model (VGPT), text-2-text model (GPT-3.5), and vision-language model
> (CLIP). During our initial submission, Video-ChatGPT (VGPT) \[1\] was
> chosen as the video-2-text model due to its competitive performance when
> compared to Video-Chat and VideoLLaMA. However, the recent introduction
> of Video-LLaVA \[2\], *a model released this week*, has set a new
> state-of-the-art (SOTA) in zero-shot question-answer and video-based
> generative benchmarks. We performed an experiment where we replaced VGPT
> with Video-LLaVA and generated video-textual descriptions for HMDB-51
> and UCF-101 benchmarks. Conducting a single inference (as opposed to the
> 10 inferences with VGPT) without any filtering applied (as now we have
> only one description per video), we observed a 0.21% and 0.75%
> improvement on previous scores for HMDB-51 and UCF-101 benchmarks,
> respectively, as shown in the table below. The adoption of Video-LLaVA
> over VGPT reduced the number of required inferences from 10 to 1,
> leading to a significant decrease in computational cost. Additionally,
> the per-video inference time for Video-LLaVA is approximately 6 times
> less than that of VGPT, with 3.79s per video, while, for reference, the
> inference time of the CLIP vision encoder is 0.74s per video. We
> anticipate that ongoing advancements in multi-model methods will further
> reduce the inference cost of VideoPrompter. We will include the updated
> results with Video-LLaVA on all benchmarks and models in our final
> manuscript.
> |         **Dataset**          | **Baseline (CLIP)** | **Video-Prompter (VGPT)** | **Video-Prompter (LLaVA)** |
> |:---------------------------:|:-------------------:|:--------------------------:|:---------------------------:|
> |           HMDB-51            |         37.5        |           52.51            |          **52.72**          |
> |           UCF-101            |         61.72       |           73.88            |          **74.63**          |
>
>
> **Q2: *Except for CLIP, the proposed approach shows relatively limited
> performance gain in other video-based models.***
>
> Please note that the large gain on the vanilla-CLIP can be linked to the
> fact that its not adopted for videos and when its used with our proposed
> framework, without any further training, the VideoPrompter brings
> significant boost in the performance. The video-based models such as
> ViFi-CLIP are already trained on Kinectics-400 and are adopted to videos
> to some extent. These models when combined with VideoPrompter increase
> the performance by 6.12%, 4.27%, and 5.3% for ViFi-CLIP, AIM, and
> ActionCLIP respectively on HMDB-51 benchmark and similar trend is seen
> on other benchmarks as shown in Table-2 and Table-6. Kindly, considering
> that no further training is performed, and the computational cost has
> been substantially reduced after Video-LLaVA, these score increments can
> be considered significant.
>
> **Q3: *The configurations of adopted pre-trained models (AIM,
> ActionCLIP, ...) remain unclear. For AIM, do the authors directly remove
> the classification layers?***
>
> Kindly note that all of the pre-trained models used in this study
> (ViFi-CLIP, AIM, and ActionCLIP) are pre-trained on Kinectics-400.
> Secondly, the original AIM model removed the text classifier from the
> vanilla CLIP and only trained the visual encoder. We removed the
> classification layers from AIM and used a text classifier from the
> vanilla CLIP with the AIM-visual encoder to get zero-shot results. These
> points have been made more clear in the original paper, as highlighted
> in red.
>
> \[1\] Maaz, et al. \"Video-ChatGPT: Towards Detailed Video Understanding
> via Large Vision and Language Models.\" arXiv.
>
> \[2\] Lin, et al. \"Video-LLaVA: Learning United Visual Representation
> by Alignment Before Projection.\" arxiv.

---

### Official Review · Reviewer_ypLm · 2023-10-31

**Soundness:** 3 good
**Presentation:** 4 excellent
**Contribution:** 3 good
**Rating:** 6
**Confidence:** 5

**Summary:**

This paper introduces a novel framework for zero-shot video understanding. The proposed framework, named VideoPromoter, is built by enhancing the visual features as well as the class representations. Experimental results indicate that the proposed method could improve the zero-shot performance of various VLMs across multiple tasks.

**Strengths:**

1.	This paper studies an important problem of adapting pre-trained vision-language models to downstream tasks in zero-shot settings.
2.	The introduced method is lucid and holds promise for extension across a wide range of VLMs.
3.	The experimental results look good. VideoPrompter is able to increase the zero-shot performance of VLMs across multiple tasks.
4.	The paper is well-presented.

**Weaknesses:**

1.	The efficiency of VideoPrompter hasn't been thoroughly examined. Given that VideoPrompter appears to require generating 10 times the number of samples and the use of an additional text-to-video model, it could substantially raise the inference costs, both for evaluating existing VLMs and in practical applications.
2.	The selection of Video-ChatGPT as the video-to-text model seems arbitrary. Alternative models, such as Video-LLaMA [A], should be considered and discussed.
3.	An ablation study on the video-specific language descriptors is missing.

[A] Zhang, H., Li, X., & Bing, L. Video-llama: An instruction-tuned audio-visual language model for video understanding. arXiv preprint arXiv:2306.02858.

**Questions:**

See weakness.

---

> ### Author Response · Authors · 2023-11-23
> **Response to Reviewer ypLm**
>
> Thank you for your detailed review, below we address the key concerns.
>
> **Q1: *The efficiency of VideoPrompter hasn't been thoroughly examined.
> It could substantially raise the inference costs.***
>
> Kindly note that the VideoPrompter offers a flexible framework, allowing
> for the substitution of various modules, including the video-2-text
> model (VGPT), text-2-text model (GPT-3.5), and vision-language model
> (CLIP). During our initial submission, Video-ChatGPT (VGPT) \[1\] was
> chosen as the video-2-text model due to its competitive performance when
> compared to Video-Chat and VideoLLaMA. However, the recent introduction
> of Video-LLaVA \[2\], *a model released this week*, has set a new
> state-of-the-art (SOTA) in zero-shot question-answer and video-based
> generative benchmarks. We performed an experiment where we replaced VGPT
> with Video-LLaVA and generated video-textual descriptions for HMDB-51
> and UCF-101 benchmarks. Conducting a single inference (as opposed to the
> 10 inferences with VGPT) without any filtering applied (as now we have
> only one description per video), we observed a 0.21% and 0.75%
> improvement on previous scores for HMDB-51 and UCF-101 benchmarks,
> respectively, as shown in the table below. The adoption of Video-LLaVA
> over VGPT reduced the number of required inferences from 10 to 1,
> leading to a significant decrease in computational cost. Additionally,
> the per-video inference time for Video-LLaVA is approximately 6 times
> less than that of VGPT, with 3.79s per video, while, for reference, the
> inference time of the CLIP vision encoder is 0.74s per video. We
> anticipate that ongoing advancements in multi-model methods will further
> reduce the inference cost of VideoPrompter. We will include the updated
> results with Video-LLaVA on all benchmarks and models in our final
> manuscript.
>
> |         **Dataset**          | **Baseline (CLIP)** | **Video-Prompter (VGPT)** | **Video-Prompter (LLaVA)** |
> |:---------------------------:|:-------------------:|:--------------------------:|:---------------------------:|
> |           HMDB-51            |         37.5        |           52.51            |          **52.72**          |
> |           UCF-101            |         61.72       |           73.88            |          **74.63**          |
>
> **Q2: *The selection of Video-ChatGPT? Alternative models should be
> considered and discussed.***
>
> Please note that our choice of video-to-text generative model is defined
> by the competitive performance \[1\] *(at the time of submission)* of
> Video-ChatGPT compared to Video-Chat and VideoLLaMA. However, the recent
> introduction of Video-LLaVA \[2\], *a model released this week*, has set
> a new state-of-the-art (SOTA) in zero-shot question-answer and
> video-based generative benchmarks. We used it as an alternative model
> and show the results below for HMDB-51 and UCF-101 benchmarks. Note
> that, for Video-LLaVA single inference is performed (as opposed to the
> 10 inferences with VGPT) and no filtration is applied (as now we have
> only one description per video). Thank you for suggesting this analysis.
> We will include this ablation in our final manuscript.
>
> |         **Dataset**          | **Baseline (CLIP)** | **Video-Prompter (VGPT)** | **Video-Prompter (LLaVA)** |
> |:---------------------------:|:-------------------:|:--------------------------:|:---------------------------:|
> |           HMDB-51            |         37.5        |           52.51            |          **52.72**          |
> |           UCF-101            |         61.72       |           73.88            |          **74.63**          |
>
>
> **Q3: *An ablation study on the video-specific language descriptors is
> missing.***
>
> As recommended, detailed ablation on language descriptors is provided
> below. We individually analyze the following components: 1)
> class-attributes, 2) class-descriptions, and 3) action-context.
> Moreover, we also analyzed various combinations of these components such
> as 1) class-attributes + class-descriptions, 2) class-attributes +
> class-descriptions + action-context. Thank you for suggesting this
> analysis. We will include this ablation in our final manuscript.
>
> | **Dataset** | **CLIP** | **Att** | **Des** | **Att + Des** | **Att + Des + Action** |
> |:-----------:|:--------:|:-------:|:-------:|:-------------:|:------------------------:|
> | HMDB-51     |   37.5   |  39.50  |  43.35  |     46.11     |       **48.57**          |
> | UCF-101     |  61.72   |  68.88  |  65.87  |     71.61     |       **72.07**          |
> |    SSv2     |   2.72   |   2.81  |   3.19  |   **3.34**    |            --              |
>
>
> \[1\] Maaz, et al. \"Video-ChatGPT: Towards Detailed Video Understanding
> via Large Vision and Language Models.\" arXiv.
>
> \[2\] Lin, et al. \"Video-LLaVA: Learning United Visual Representation
> by Alignment Before Projection.\" arxiv.

---

### Official Review · Reviewer_1vg9 · 2023-10-31

**Soundness:** 2 fair
**Presentation:** 3 good
**Contribution:** 2 fair
**Rating:** 5
**Confidence:** 4

**Summary:**

This paper proposes to ensemble multiple large foundation models to enhance the zero-shot inference performance on video understanding tasks (namely VideoPrompter), including video action recognition, video-to-text and text-to-video retrieval, and time-sensitive (before/after) video tasks. The main architecture is based on CLIP, where classification can be performed by ranking the cosine similarity between visual and text representations, and the main idea is to enrich both the video and text embeddings. For the video part, the authors employ Video-Chat GPT (VGPT) (Maaz et al., 2023) to extract the text description of the query video and convert it into a video-to-text embedding with the text encoder in CLIP. The video-to-text embedding is then ensembled with the visual embedding encoded by the original CLIP visual encoder as the final visual embedding. For the text part, they prompt GPT-3.5 to rephrase the class names with parent context, language attributes, and language descriptions. All the descriptions are ensembled to generate the final text embedding. Experiments show that VideoPrompter can improve over plain zero-shot inference performance with CLIP and its variants.

**Strengths:**

1. The studied problem is interesting. Video understanding with large foundation models is of wide interest in the community.
2. The authors put together state-of-the-art large foundation models and improve the zero-shot inference performance on video understanding tasks.

**Weaknesses:**

1. The idea of generating more descriptions for class names and using high-level context is not new in prompting large foundation models (e.g. the prior works cited in this paper). This is model ensembling for enhancing zero-shot performance. Can the authors justify the main novelty of this paper?
2. VGPT is used to generate the text description of the query video, and which is then converted to an image-like text embedding. Why not just prompting VGPT for the downstream applications (e.g. action classification)? Comparison to this baseline is an important justification to the proposed method.
3. Several components are added to the solution, while the ablations are not sound enough. For example, how important are the three description types (parent context, language attributes, and language descriptions)?
4. The claim for the comparison to CUPL (Pratt et al., 2022) is not very clear (section 3.1.4). The authors claim that VideoPrompter only requires 3 text descriptions instead of 50 descriptions adopted in CUPL. However, VideoPrompter adopts a VGPT model while CUPL does not. Is using VGPT a better choice in terms of the cost?
5. The paper criticizes prior work that “these methods require access to the true distribution of the target task, which can be prohibitive in test-time adaptation and data-scarce environments”. However, the proposed method optimizes the selection of hyperparameters (e.g. temperature) directly on the target dataset (see Figure 3).
6. The high-level action context is restricted to a tree-type relation. However, some child classes may belong to multiple parent concepts. For example, “surfing” can belong to both “playing sports” and “water activities”.

**Questions:**

My questions are listed in the weakness section.

---

> ### Author Response · Authors · 2023-11-23
> **Response to Reviewer 1vg9**
>
> Thank you for your detailed review, below we address the key concerns.
>
> **Q1: *Can the authors justify the main novelty of this paper?***
>
> Kindly note that our work is the first one to study the role of
> video-to-text models for video understanding and highlights how video descriptions can
> further enhance visual representations. So far, the existing methods
> \[1,2\] are only limited to text-to-text generative models for enhancing
> the text classifier representations only. We summarize our contributions
> as follows:
>
> 1.  So far, only text-to-text generative models (e.g. GPT) are used to
>     enhance the representations of the text classifier. Our work
>     discusses the impact of having video-to-text generative models (e.g.
>     VGPT) to enrich the visual representations as well, in addition to
>     only text-based models (Section 2.2).
>
> 2.  We propose a novel way of querying LLMs called action context
>     (Section 2.3.2). Action context can classify semantically close
>     videos under a single high-level context (category). In the related
>     work, we discuss the limitations of the existing methods \[3,4\] and
>     discuss how these methods are only applicable to image benchmarks
>     and are unsuited for video benchmarks where the classes can be
>     fine-grained as well can have a diverse range.
>
> 3.  We test our framework in three different video settings namely:
>     action recognition, video-to-text, and text-to-video retrieval, and
>     time-aware video tasks. We show results with 4 different models on 7
>     benchmarks.
>
> **Q2: *Why not just prompting VGPT for the downstream applications?***
>
> As recommended, we designed an experiment where only VGPT is directly
> used to classify the input video. We found this setting to degrade the
> results by a large margin. We relate this to the training of VGPT as it
> is trained on generating long contexts about the input, rather than
> specific categories \[5\]. We also present a more extended setting,
> where the descriptions from VGPT are given to GPT-3.5 to predict the
> class name. Here, the GPT-3.5 is also provided by the list of class
> names. It can be seen that VideoPrompter outperforms both of
> the aforementioned settings which endorses its design choice. Thank you
> for suggesting this analysis. We will include this ablation in our final
> manuscript.
> |     **Datasets**     | **Vanilla-CLIP** | **VGPT only** | **VGPT+GPT** | **VideoPrompter** |
> |:--------------------:|:----------------:|:-------------:|:------------:|:------------------:|
> |       HMDB-51        |       37.5       |      16.43    |      46.0      |      **52.51**     |
> |       UCF-101        |       61.5       |      39.21    |      68.0      |      **73.88**     |
>
> **Q3: *The comparison with CUPL.***
>
> We regret any confusion, it's important to note that a direct comparison
> with CUPL may not be straightforward, as it can be viewed as a
> complementary approach to our method. As discussed in \[3\], the
> performance of such methods benefits from averaging over a large number
> of descriptors. When we combined CUPL with
> VideoPrompter i.e. weights of language descriptors are averaged,  performance
> increased by  0.12% for UCF-101. We will further update
> this discussion in our final manuscript.
>
> **Q4: *The proposed method optimizes the selection of hyperparameters
> directly on the target dataset.***
>
> Kindly note that Figure 3 is mainly added to show an ablation analysis.
> The selection of hyper-parameters in our work follows the standard
> settings in recent studies \[1,6\]. For instance, \[1\] includes a
> thorough discussion of how a high value of temperature leads to more
> diverse descriptions. And \[6\] discusses a similar filtering method.
>
> **Q5: *The Action context is restricted to a tree-type relation.***
>
> We thank the reviewer for the pointer, this indeed could be true for
> some benchmarks as some classes may overlap. We designed an experiment
> where we prompted the LLM and explicitly mentioned that one child class
> can be assigned to multiple parent classes. We found such cases to be
> relatively small, e.g. we didn't find any such case for HMDB-51 and only
> a few for UCF-101, which increased our performance by 0.06% on top of
> the previous score. We will include the updated results on all
> benchmarks and models in our final manuscript.
>
> \[1\] Pratt, et al. \"What does a platypus look like? generating
> customized prompts for zero-shot image classification.\" ICCV.
>
> \[2\] Menon. \"Visual classification via description from large language
> models.\" arXiv.
>
> \[3\] Roth, et al. \"Waffling around for Performance: Visual
> Classification with Random Words and Broad Concepts.\" arXiv.
>
> \[4\] Novack, et al. \"Chils: Zero-shot image classification with
> hierarchical label sets.\" ICML.
>
> \[5\] Maaz, et al. \"Video-ChatGPT: Towards Detailed Video Understanding
> via Large Vision and Language Models.\" arXiv.
>
> \[6\] Li, et al.\"Blip: Bootstrapping language-image pre-training for
> unified vision-language understanding and generation.\" ICML.

---

### Official Review · Reviewer_8fLg · 2023-11-01

**Soundness:** 3 good
**Presentation:** 2 fair
**Contribution:** 3 good
**Rating:** 5
**Confidence:** 3

**Summary:**

This paper proposes a framework for zero-shot video understanding by using various foundation models including VLMs, i.e., CLIP, LLMs, i.e., GPT, and Video-to-Text model, i.e., VGPT. Experiments are conducted on three different problem settings and showing good results. Ablations are thorough and enough to justify the framework design choices. Written presentation is fair, but could be improved.

**Strengths:**

- The paper presents a set of experiments on various problem settings: action recognition, video-to-text and text-to-video retrieval, time-sensitive tasks and on different datasets.
- The ablations are solid and thorough.
- Experiments show strong improvement w.r.t baselines.

**Weaknesses:**

- Since at least 3 foundation models have been used (CLIP, GPT, VGPT), how do we know if those models are trained with examples overlapped with the downstream datasets (e.g., HMDB-51, UCF101, SSv2, K400, MSR-VTT, Charades).

- The novelty seems moderate if not low. As the paper mentions the main contributions are 1) introducing video-to-text to enhance visual embeddings and 2) applications to videos.

- The written presentation could be further improved:
     1) section 2.1 could be renamed to "Overview" and try to capture the big picture of the framework. The author(s) can refer back to Fig. 1 for the big picture (in the current flow of presentation, there is no big picture and it flows in with overwhelming many details and notations). Then sections 2.2 and 2.3 can be further followed up from 2.1 to provide detailed of components.
     2) table 6 is presented in page 7, yet never been referred from the text?

**Questions:**

- My main concerns are the leaking examples from downstream datasets to foundation models.

---

> ### Author Response · Authors · 2023-11-23
> **Response to Reviewer 8fLg**
>
> Thank you for your detailed review, below we address the key concerns.
>
> **Q1: *Leaking examples from downstream datasets to foundation
> models*.**
>
> Kindly note that the CLIP is trained on 400M image-text pairs and the
> VGPT employs video-text pairs and is trained on a subset of
> ActivityNet-200. While the GPT is only trained on the text data. To
> investigate any overlapping with the downstream datasets we design the
> following experiment and present results on HMDB-51 and UCF-101
> benchmarks:
>
> 1.  We show zero-shot results of vanilla-CLIP on HMDB-51 and UCF-101
>     benchmarks to understand the extent of knowledge the CLIP model
>     contains about these datasets.
>
> 2.  Second, to understand any data overlap of VGPT, we only employ VGPT
>     directly to classify the input video. We found this setting to
>     degrade the results by a large margin. We relate this to the
>     training of VGPT as it is trained on generating long contexts about
>     the input, rather than specific categories \[1\].
>
> Kindly, from the table below, it can be seen that the individual
> performance of these modules (vanilla-CLIP and VGPT) is substantially
> low, which shows minimum data overlap. Thank you for suggesting this
> analysis. We will include this ablation in our final manuscript.
>
> |   **Datasets**   | **CLIP** | **VGPT only** | **VideoPrompter** |
> |:----------------:|:--------:|:-------------:|:------------------:|
> |     HMDB-51      |   37.5   |     16.43     |      **52.51**     |
> |     UCF-101      |   61.5   |     39.21     |      **73.88**     |
>
>
> **Q2: *The novelty seems moderate if not low.***
>
> Kindly note that our work is the first one to study the role of
> video-to-text models for video understanding. Our work highlights how
> the recent advances in the video-to-text models can be used in the
> general video-understanding pipeline, and how video descriptions can
> further enhance the visual representations. So far, the existing methods
> \[2,3\] are only limited to text-to-text generative models for enhancing
> the text classifier representations only. We summarize our contributions
> as follows:
>
> 1.  So far, only text-to-text generative models (e.g. GPT) are used to
>     enhance the representations of the text-classifier. Our work
>     discusses the impact of having video-to-text generative models (e.g.
>     VGPT) to enrich the visual representations as well, in addition to
>     only text-based models (Section 2.2).
>
> 2.  We propose a novel way of querying LLMs called action context
>     (Section 2.3.2). Action context can classify semantically close
>     videos under a single high-level context (category). In the related
>     work, we discuss the limitations of the existing methods \[4,5\] and
>     discuss how these methods are only applicable to image benchmarks
>     and are unsuited for video benchmarks where the classes are
>     fine-grained as well as have diverse ranges.
>
> 3.  We test our framework in three different video settings namely:
>     action recognition, video-to-text, and text-to-video retrieval, and
>     time-aware video tasks. We show results on 7 benchmarks including
>     HMDB-51, UCF-101, SSv2, K400, MSRVTT, Charades, and the time-aware
>     synthetic dataset. Moreover, as our framework offers plug and play
>     module, we show results with various models such as CLIP, ViFi-CLIP,
>     Action-CLIP, and AIM.
>
> **Q3 (a): *The written presentation could be further improved.***
>
> Please note that, sections 2.1, 2.2, and 2.3 have been updated. Now, for
> more clarity we have re-named section 2.2 from video-specific language
> descriptors to class-specific language descriptors. The revised text is
> highlighted in red.
>
> **Q3(b): "*table 6 is presented in page 7, yet never been referred from
> the text?***
>
> Kindly note that table 6 is referred in video action recognition section
> on page 6.
>
>
> \[1\] Maaz, et al. \"Video-ChatGPT: Towards Detailed Video Understanding
> via Large Vision and Language Models.\" arXiv.
>
> \[2\] Pratt, et al. \"What does a platypus look like? generating
> customized prompts for zero-shot image classification.\" ICCV.
>
> \[3\] Menon. \"Visual classification via description from large language
> models.\" arXiv.
>
> \[4\] Roth, et al. \"Waffling around for Performance: Visual
> Classification with Random Words and Broad Concepts.\" arXiv.
>
> \[5\] Novack, et al. \"Chils: Zero-shot image classification with
> hierarchical label sets.\" ICML.

---

### Author Response · Authors · 2023-11-23
**Main Response Comment**

We thank all the reviewers (\(8fLg\), \(1vg9\), \(ypLm\), \(FnrC\)) for their thoughtful reviews and appreciate the detailed comments to improve our work.

**Reviewer-8fLg:** "*The ablations are solid and thorough. Experiments show strong improvement w.r.t baselines*".

**Reviewer-1vg9:** "*The studied problem is interesting*".

**Reviewer-ypLm:** "*This paper studies an important problem. The experimental results look good. The paper is well-presented*".

**Reviewer-FnrC:** "*The experiments and ablation studies are conducted comprehensively. Employing the video-to-text model (not limited to VGPT and caption models) is novel and worthy to explore in the video field. The manuscript provides a detailed explanation and examples of prompting the GPT to refine the simple textual label, which in turn enhances reproducibility*".


**Our codes will be publicly released**. We summarize the salient features of the proposed approach below:

- Our work is the first one to study the role of video-to-text models for video understanding.
- Our work discusses the impact of having video-to-text generative models (e.g., VGPT) to enrich the visual representations, in addition to only text-based models (e.g., GPT) for enhancing the text-classifier representations.
- We propose a novel way of querying LLMs referred to as action context. It can classify the semantically closed videos under a single high-level context (category).
- We test our framework in three different video settings, namely: action recognition, video-to-text, and text-to-video retrieval, and time-aware video tasks. We show results with 4 different models on 7 benchmarks.

---

### Meta-Review · Area_Chair_jJWM · 2023-12-09

**Metareview:**

This paper received overall borderline ratings (5, 5, 6, 5). The reviewers generally acknowledged that the studied problem of leveraging large foundation models for zero-shot video understanding is timely and interesting, and that the experiments are comprehensive with thorough ablations. The major concerns included 1) the extra computational complexity brought by the approach and 2) the limited novelty, as the idea of prompting LLMs for more text supervision has been well explored already.

The authors provided a rebuttal with new ablations and quantitative analysis. The reviewers generally seem to be satisfied with the rebuttal, although their ratings still remain borderline (leaning rejection). In regards to the extra compute concern, the new results with Video-LLaVA show that the 10x inference requirement isn't strictly necessary, which effectively addresses the issue. On the novelty concern, the authors asserted that this work is the first to explore video-to-text for zero-shot video understanding, and asserted this as their core novelty. However, prior work, such as [A], has already explored this direction, leveraging a video-conditioned LLM to "narrate" video clips and using them as weak language supervision to enhance video understanding. This paper was not referenced and discussed in this submission. Considering that the authors emphasized the video-to-text component as the core novelty, it is suggested that further clarification and discussion be provided to address the novelty concerns. The submission, in its current form, may benefit from revisions to better substantiate the uniqueness of the proposed approach.

[A] Yue Zhao et al. "Learning Video Representations from Large Language Models." CVPR 2023.

**Justification For Why Not Higher Score:**

The novelty concern as described above.

**Justification For Why Not Lower Score:**

N/A

---

### Decision · Program_Chairs · 2024-01-16

Reject